# Joint Gaussian Mixture Model for Versatile Deep Visual Model Explanation

## Abstract

Post-hoc explanations of deep neural networks improve human understanding on the learned representations, decision-making process and uncertainty of the model with faithfulness. Explaining deep convolutional neural networks (DCNN) is especially challenging, due to the high dimensionality of deep features and the complexity of model inference. Most post-hoc explaining methods serve a single form of explanation, restricting the diversity and consistency of the explanation. This paper proposes joint Gaussian mixture model (JGMM), a probabilistic model jointly models inter-layer deep features and produces faithful and consistent post-hoc explanations. JGMM explains deep features by Gaussian mixture model and inter-layer deep feature relations by posterior distribution on the latent component variables. JGMM enables a versatile explaining framework that unifies interpretable proxy model, global or local explanatory example generation or mining. Experiments are performed on various DCNN image classifiers in comparison with other explaining methods. It shows that JGMM can efficiently produce versatile, consistent, faithful and understandable explanations.

## 1 Introduction

Deep convolutional neural networks (DCNN) is a powerful type of machine learning model for visual recognition tasks. The key reasons give rise to the power of DCNN include the expressive visual representations and the decision-making mechanism encoded in massive trainable convolution parameters. However, increasing model complexity incurs heavier burden for human to understand learned representations and decision making of the model. The high dimensionality and entanglement of deep features and the complexity of neural network inference are often considered the main hindrance to explaining black-box DCNN.

A recent proliferation of studies in post-hoc DCNN explainability show several effective and practical DCNN explaining methods.

Proxy models are interpretable models (e.g. decision trees and linear models) that has approximate decision-making behavior as the black-box model inference. Proxy models have inference process that can be intuitively understood by human, such as LIME (proposed by Ribeiro et al. (2016)), a linear classifier as a local proxy model. To make sure the proxy model is an accurate surrogate, its faithfulness should be tested. The proxy model's predictions on unseen examples should be close to the black-box model's prediction, even if different from the ground truth.

The intermediate representations of DCNN are usually explained globally (not associated with a specific data point) by explanatory examples and association with semantic concepts. Prototypes, criticisms and influential examples are common types of global explanatory examples. Prototypes are representative examples of a certain pattern of deep features, illustrating a learned visual concept of the model. In contrast, criticisms (proposed by Kim et al. (2016)) are examples not well-represented in the deep representations, i.e. outliers of the deep features, revealing the flaw of the learned representations. Influential examples are hard examples for the model training, having more influence on the final decision boundary than others. From a post-hoc view, influential examples lie close to the decision boundary. An example can be both influential and representative (or unrepresentative).

Local explanations are based on a specific query example, showing how the change of the query example features will affect the model prediction. Counterfactual examples offer an actionable re-

course for the model decision. Counterfactual examples answer the 'what if' questions by minimal change of query example features and resulting different model decision against the query. Counterfactual examples reveal the sensitive features for the query. Semi-factual examples, in contrast, aim to answer the 'even if' question. Semi-factual examples have significant change on a certain feature(s) from the query, but both have the same model prediction. Semi-factual examples reveal the insensitive features of the query. For example, Kenny & Keane (2021) proposes a method to generate counterfactual and semi-factual examples from one system.

Most DCNN explaining methods are single-purpose systems. Employing different explaining methods is possible to give diverse explanations, but it's not guaranteed that different explanations are compatible and consistent with each other. For example, a counterfactual example generation system may suggest that the model is sensitive to a certain feature; but a global explaining method, such as a proxy model, may have conflicts with the former explanation. There is no hard rule to determine which explanation is correct or more understandable. Thus a generic explaining framework that enables various and consistent explaining forms has important value for the explainability of DCNN.

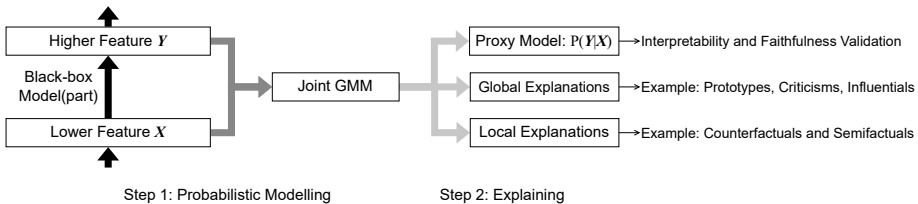

Figure 1: Proposed probabilistic framework for a versatile explaining method. The intermediate features ($\mathbf{X}$ and $\mathbf{Y}$) are jointly modelled by a probabilistic model joint Gaussian mixture model (JGMM).

We propose a probabilistic framework for a versatile explaining method for DCNN. Figure 1 demonstrates the pipeline of the framework and the enabled explanation forms. The lower and the higher features are intermediate representations of DCNN from two different layers. In the computational graph of DCNN, the higher feature is dependent on the lower feature and the black-box model (part of the DCNN) between them. In an image classification setting, the higher feature can be the classification probabilistic prediction of the black-box model, and the lower features can be the input of the DCNN, i.e. the raw image. The goal of the framework is to explain the learned representations of higher/lower features as well as the black-box model (either the whole DCNN or a part of DCNN) between them.

To serve this purpose, we propose joint Gaussian mixture model (JGMM) to jointly model the distribution of lower/higher features and produce a validatable proxy model and example-based global/local explanations. JGMM is a probabilistic model based on GMM. JGMM learns two Gaussian mixture model (GMM) respectively for the lower features $\mathbf{X}$ and higher features $\mathbf{Y}$. The latent categorical variables of both GMMs are connected by an estimated posterior probability matrix. JGMM is introduced in section 3 in details.

Compared with single purpose explaining methods, the proposed JGMM-based versatile explaining method has two advantages: (1) various forms of model explanations are efficiently produced from one framework; (2) the consistency among different explanations is guaranteed, as they are computed from a common probabilistic model. The proposed explaining method is evaluated with various DCNN models and benchmarks in comparison with other explaining methods in section 4. The experiments show that JGMM can efficiently produce versatile, consistent, faithful and understandable explanations.

## 2 RELATED WORKS

Proxy models are widely leveraged method to produce model-agnostic explanations. Local interpretable model-agnostic explanations (LIME) by Ribeiro et al. (2016) is a typical method that learns a linear classifier by sampling from data points from a local region and the black-box classier. For

deep networks, Zilke et al. (2016) propose to convert model parameters to a hard rule-set but keep it a computational approximation. To explain the visual concepts encoded in DCNN and how the decision is made, Zhang et al. (2018) and Zhang et al. (2019) respectively design explanatory graphs and decision trees to explain the model inference.

Pixel attributing, or saliency mapping, is another useful technique to explain DCNN. Compared with traditional feature attributing of machine learning models on tabular data, pixel attributing is specialized to the hierarchical structure of DCNN and visualizes the attention of the model. Methods like CAM by Zhou et al. (2016), Integrated Gradients by Sundararajan et al. (2017), DeepLIFT ny Shrikumar et al. (2017), Grad-CAM ny Selvaraju et al. (2017) utilize different heuristics to back-propagate the feature importance from deep layers to shallow layers.

Example-based explaining is a less explored field for deep visual model explanation. Studies like Goyal et al. (2019), Kenny & Keane (2021) and Black et al. (2021) generate counterfactual examples by simultaneous minimization of the feature distance after the query and maximization of model decision change. Prototypes, the representative examples of a feature space cluster, are usually employed to explain a certain understandable visual concept learned by the model. For example, Nauta et al. (2021) propose ProtoPNet to mine prototypical image patches and explain object parts. Semi-factual examples (e.g. Kenny & Keane (2021)), criticisms (e.g. Kim et al. (2016)) and influential examples (e.g. Koh & Liang (2017)) can also help explaining the distribution of deep model features.

## 3 METHOD

### 3.1 JOINT GAUSSIAN MIXTURE MODEL

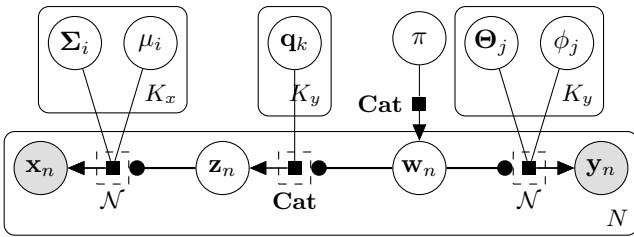

Figure 2: Graphical model of joint Gaussian mixture model (JGMM) on lower feature $\mathbf{X}$ and higher feature $\mathbf{Y}$.

We propose joint Gaussian mixture model, a probabilistic model base on GMM to estimate the posterior distribution connecting the latent variables behind the deep features. Figure 2 illustrates the model in plate notation. The model includes three parts: the GMM on observation $\mathbf{Y}$, the conditioned GMM on observation $\mathbf{X}$, and the probabilistic model between discrete latent variables $z$ and $w$.

The model on observation $\mathbf{Y}$ resembles the classical GMM. It includes the observed $D_y$-dimension real-value higher features $\mathbf{Y} \sim \mathcal{N}(\phi_w, \Theta_w)$. Data point $\mathbf{y}_n$ is the $n$-th of $N$ sampled higher features. The latent categorical variable $w \in \{1, ..., K_y\}$ decides which Gaussian component is $\mathbf{Y}$ generated from. Hyper-parameter $K_y$ is the number of mixture components. Prior probability $\pi$ is a $K_y$-D vector which represents the weight of each $K_y$ mixture component.

The model on observation $\mathbf{X}$ is also a Gaussian mixture model. The observed $D_x$-dimension lower features $\mathbf{X}$ is generated by the $z$-th Gaussian component: $\mathbf{X} \sim \mathcal{N}(\mu_z, \Sigma_z)$. Latent categorical variable $z \in \{1, ..., K_x\}$ has hyper-parameter $K_x$, which is the number of mixture components. However, the key difference of our model from a traditional GMM is that the prior probability of $z$ is dependent on posterior probability $p(z|w)$ parameterized by $\mathbf{Q}$.

The connection between latent categorical variables $w$ and $z$ is modeled by the posterior probability matrix $\mathbf{Q} \in \mathcal{R}^{K_x \times K_y}$. The entry $\mathbf{Q}_{i,j}$ is the conditional distribution of $z$, i.e. $p(z = i|w = j)$. In figure 2, $\mathbf{q}_k$ is the $k$-th column vector of $\mathbf{Q}$, i.e. the distribution of $z$ conditioned on $w = k$, whose sum is 1.

By the chain rule of Bayesian inference, the likelihood function of JGMM is($G$ is Gaussian distribution density function):

$$p(\mathbf{x}, \mathbf{y}) = \sum_{i=1}^{K_x} \sum_{j=1}^{K_y} p(\mathbf{x}|z=i)p(\mathbf{y}|w=j)p(z=i|w=j)p(w=j)$$

$$= \sum_{i=1}^{K_x} \sum_{j=1}^{K_y} G(x; \mu_i, \Sigma_i)G(y; \phi_j, \Theta_j)\mathbf{Q}_{i,j}\pi_j \tag{1}$$

The maximum likelihood estimation (MLE) problem of JGMM parameters can be solved by expectation-maximization (EM) algorithm. The optimization algorithm can be straightforwardly derived from the EM algorithm framework, whose details are described in the appendix section A. The complexity of the optimization is similar to separately learning two GMM for $\mathbf{X}$ and $\mathbf{Y}$.

Equation 1 shows that the trained JGMM is still a GMM with $(K_x K_y)$ components, preserving the desired properties of GMM. Discussions below use the following notations for simplicity. For a given observation $\mathbf{x}$ and $\mathbf{y}$, $g_x \in \mathbb{R}^{(K_x \times 1)}$ and $g_y \in \mathbb{R}^{(K_y \times 1)}$ are probability density on each component, and $\Psi \in \mathbb{R}^{(K_x \times K_y)}$ is the conditional distribution of $\mathbf{w}$:

$$[g_x]_i = p(x|\hat{\mu}_i, \hat{\mathbf{\Sigma}}_i), [g_y]_j = p(y|\hat{\mu}_j, \hat{\Theta}_j), \hat{\lambda}_i = p(\mathbf{z}=i) = \hat{\mathbf{Q}}_{i,*}\hat{\pi}$$

$$\hat{\mathbf{\Psi}}_{i,j} = p(\mathbf{w}=j|\mathbf{z}=i) = \frac{\hat{\mathbf{Q}}_{i,j}\hat{\pi}_j}{\sum_{k=1}^{K_y} \hat{\mathbf{Q}}_{i,k}\hat{\pi}_k}, i=1,...,K_x, j=1,...,K_y \tag{2}$$

where $\hat{\mathbf{Q}}, \hat{\mu}, \hat{\mathbf{\Sigma}}, \hat{\phi}, \hat{\mathbf{\Theta}}$ and $\hat{\pi}$ are parameters estimated by MLE. $\hat{\mathbf{Q}}_{i,*}$ is the $i$-th row vector of $\hat{\mathbf{Q}}$.

Trained JGMM can work as a bi-directional Bayesian classifier to predict latent variable $\mathbf{z}$ and $\mathbf{w}$. Conditioned on a known $\mathbf{x}$ or $\mathbf{y}$, the distribution of the other feature's latent component variable $\mathbf{w}$ or $\mathbf{z}$ is:

$$p(\mathbf{w}=j|\mathbf{x}) = \frac{(g_x^T \hat{\mathbf{Q}}_{*,j})\hat{\pi}_j}{(g_x^T \hat{\mathbf{Q}}\hat{\pi})}, p(\mathbf{z}=i|\mathbf{y}) = \frac{(\hat{\mathbf{\Psi}}_{i,*}g_y)\hat{\lambda}_i}{(\hat{\lambda}\hat{\mathbf{\Psi}}g_y)} \tag{3}$$

Posterior probability in equation 3 is naturally a Bayesian classifier to predict the component category of the given example. This enables a traditional training-test evaluation manner on the proxy model to evaluate the faithfulness of modelling.

JGMM also makes a handy example generator and miner to improve human understanding on the model by examples. When the conditional distribution of $\mathbf{w}$ or $\mathbf{z}$ is computed by equation 3, JGMM on $\mathbf{y}$ or $\mathbf{x}$ degrades to a normal GMM, which is easy for sampling and interpretation.

## 3.2 MINING AND GENERATING EXPLANATORY EXAMPLES

This section introduces how to mine or generate explanatory examples (prototypes, criticisms, influential examples, counterfactuals and semi-factuals) by a trained JGMM. The example mining problem is to search for the best explanatory examples in the dataset, which requires a scoring function $f(\mathbf{x})$ to evaluate the quality. Generation problem is the sampling of data that maximize the scoring function. We respectively define the scoring function for each type of explanatory examples and introduce the way to mine or generate examples.

**Prototypes.** From a probabilistic aspect, a representative example $x$ given $w = j$ is at the center of the cluster, whose neighboring region has high cumulative probability on $p(x|w=j)$, which is a GMM. So we define the prototype scoring function of the $j$-th higher feature component as equation 4.

$$f_{\text{proto}}^{(j)}(x) = p(x|w=j) = GMM(\hat{\mu}, \hat{\mathbf{\Sigma}}, \hat{\mathbf{q}}_j) \tag{4}$$

Sampling prototypes for the $j$-th higher feature component is trivially sampling from multivariate Gaussian distribution $p(x|w=j)$.

**Criticisms.** Criticisms can be considered as outliers that is hardly represented by any prototypes. So we search for examples that are least likely sampled from any $w$, which is the opposite of the

prototype score, as equation 5.

$$f_{\text{criti}}(\mathbf{x}) = 1 - \max_j f_{\text{proto}}^{(j)}(x) \tag{5}$$

Sampling criticisms is not supported in our method, because the realistic data distribution is often very sparse compared to the feature space. Most sampled outliers are not understandable examples. We only focus on criticisms found in realistic data.

**Influential examples.** Influential examples are close to the decision boundary and can lead to model decision with low confidence. From JGMM, we can measure the influence of an example by the entropy of the posterior probability of $w$ conditioned on $\mathbf{x}$. Higher the entropy, higher the uncertainty and influence. The scoring function is defined as equation 6.

$$f_{\text{influ}}(\mathbf{x}) = - \sum_{j=1}^{K_y} p(w = j|\mathbf{x}) \ln p(w = j|\mathbf{x}) \tag{6}$$

Influential examples can be generated by sampling from each components of $\mathbf{X}$ and select those with largest $f_{\text{influ}}(\mathbf{x})$. Sampled influential examples, though not actually in the training data of the black-box, can be useful for exploring uncertain examples for the black-box model.

**Counterfactuals.** A counterfactual example is close to the query example $\mathbf{x}_{\text{query}}$ in the feature space but have very different posterior distribution on $w$. Therefore, the scoring function for counterfactuals is defined as the probability that $\mathbf{x}$ and $\mathbf{x}_{\text{query}}$ are sampled from the same $z$ but different $w$:

$$f_{\text{count}}(x) = \sum_{i}^{K_x} \sum_{j}^{K_y} p(w = j|z = i) p(z = i|x) p(w_{\text{query}} \neq j|z_{\text{query}} = i) p(z_{\text{query}} = i|x_{\text{query}}) \tag{7}$$

To maximize the counterfactual score, there actually exists an optimal distribution of $p(z|x)$. Let this distribution be $\alpha$, the maximization of equation 7 is:

$$\max_{\alpha} \sum_{i}^{K_x} \alpha_i (\sum_{j}^{K_y} p(w = j|z = i) p(w_{\text{query}} \neq j|z_{\text{query}} = i) p(z_{\text{query}} = i|x_{\text{query}})), s.t. \sum_{i}^{K_x} \alpha_i = 1 \tag{8}$$

Note that all terms except $\alpha$ are constant, which makes it a linear programming problem with sum constraint. Though it's still hard to derive optimal $x$, the solution still provides an efficient sampling approach. The approach to generate counterfactuals is to sample from the GMM under this optimal distribution $\alpha$ and select the best counterfactuals.

**Semi-factuals.** A semi-factual example is likely sampled from a different $z$ from the query example $\mathbf{x}_{\text{query}}$ but have probably the same posterior distribution on $w$, which is the reverse of counterfactuals. Therefore, the scoring function for counterfactuals is defined as equation 9.

$$f_{\text{semi}}(x) = \sum_{i}^{K_x} \sum_{j}^{K_y} p(w = j|z = i) p(z = i|x) p(w_{\text{query}} = j|z_{\text{query}} \neq i) p(z_{\text{query}} \neq i|x_{\text{query}}) \tag{9}$$

Similar as the counterfactuals, the maximization of semi-factual score leads to an optimal distribution of $p(z|\mathbf{x})$. We sample from the GMM of this optimal distribution and select the best semi-factuals.

### 3.3 EXTENSION FOR DCNN

The deep feature maps of DCNN usually has additional spatial dimensions apart from the feature dimension, such as the time dimension of signal data and the vertical and the horizontal dimensions of 2D images. The feature maps can be formulated with spatial encoding $s \in S$, i.e. $\mathbf{X}^{(s)}$. For example, for a feature map of width $W$ and height $H$, the spatial encoding is $S = \{(i, j)|1 \leq i \leq W, 1 \leq j \leq H\}$. In later discussions, $S$ and $T$ respectively denote the spatial dimensions of $\mathbf{X}$ and $\mathbf{Y}$.

The addition of spatial dimensions makes the latent variable relation changes from one-to-one to many-to-many, i.e. modelling each pair of $\{(s, t)|s \in S, t \in T\}$. Theoretically, JGMM can be

extended to such scenario by estimating each of $Q^{(s,t)}$, and make GMM parameters $\mu$, $\boldsymbol{\Sigma}$, $\phi$, $\boldsymbol{\Theta}$, $\pi$ shared by every $s \in S$. However, both E-step and M-step in the EM algorithm of JGMM requires enumeration at all possible latent variable assignment (see equation 13), where the combinatorial space ($K_x^{|S|} K_y^{|T|}$ elements) is so large that enumeration is infeasible.

However, most modern DCNN architectures have translation-invariant mapping of features, which means translation of lower features over spatial dimensions will cause the same translation (relative) of higher features. Also, though the relative receptive field (the region on the lower features that affects a specific position of higher feature) of $T$ on $S$ can be very large, research by Luo et al. (2016) shows that the central area (the effective receptive field) has main influence on the higher feature. Therefore we can extend the proposed probabilistic model to a one-to-many relation. Let this relative effective receptive field be $R$. $R$ is a region cast the lower feature and connected to a point on higher feature. We train $|R|$ posterior distribution matrix $\mathbf{Q}^{(r)}$ between $S$ and $T$. The size of $R$ is a hyper-parameter. Also, the parameters $\mu$ and $\boldsymbol{\Sigma}$ are shared by all positions. Figure 3 shows the graphical model of JGMM extended for deep features with spatial dimensions. The hyper-parameter $|R|$ decides how many lower feature positions will affect the higher feature. Like the setting of convolution hyper-parameters, the size of the effective receptive field should satisfy $|S| + 1 - |T| \geq |R|$.

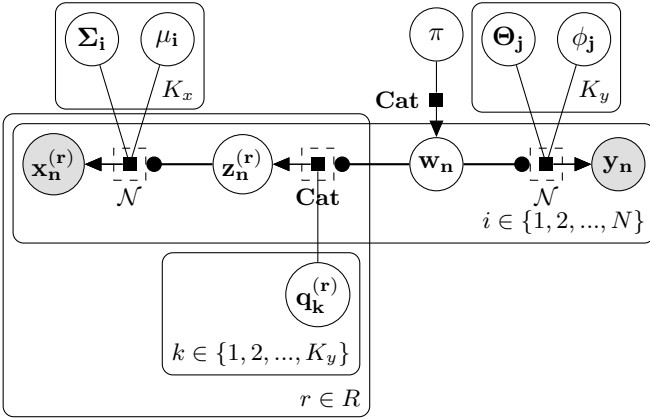

Figure 3: Extension of JGMM to relative spatial dimension $R$.

An important property of the extended JGMM is that $\mathbf{X}^{(r)}$ is conditionally independent from other $r' \in R$ on $w$:

$$p(\{\mathbf{x}^{(r)} | r \in R\} | w) = \prod_{r \in R} p(\mathbf{x}^{(r)} | w) \tag{10}$$

So the inference of the extended JGMM can be decomposed to the inference on each of the $|R|$ JGMMs:

$$p(w | \{\mathbf{x}^{(r)} | r \in R\}) = \frac{p(\{\mathbf{x}^{(r)} | r \in R\} | w) p(w)}{p(\{\mathbf{x}^{(r)} | r \in R\})} = \frac{p(w) \prod_{r \in R} p(\mathbf{x}^{(r)} | w)}{p(\{\mathbf{x}^{(r)} | r \in R\})} \tag{11}$$

Therefore, the inference and sampling methods of JGMM are directly applicable to the extended JGMM.

## 4 EXPERIMENTS

### 4.1 IMPLEMENTATION DETAILS

**JGMM implementation.** For a vanilla GMM, the covariance matrix contributes the most parameters to be estimated. To reduce the complexity of JGMM training, we whiten the lower and higher

features by PCA (principal component analysis). All PCA components are kept to make the transformation invertible and all dimensions are uncorrelated with each other empirically. Thus, training JGMM can be less complex by estimating two diagonal covariance matrices, i.e. only the diagonal entries of $\Sigma$ and $\Theta$ are estimated and others are constantly zero.

**Data splitting.** To fairly test the faithfulness of JGMM, we only use the training data of the black-box model to train JGMM and use the test data to evaluate JGMM's accuracy in predicting the black-box model's decision. The effectiveness test of counter/semi-factual example generation is also performed with the test data. But the explaining step (the generation of all kinds of explanatory examples) makes use of all data.

**Features.** For a DCNN image classifier, the classification output (without softmax) is always selected as a higher feature to explain. Because it directly links the understandable image classes with feature dimensions. For models with large input image resolution, the feature map is very large and, because JGMM sample from each position, makes the training of JGMM difficult. To balance the data size and training feasibility, the feature maps are average-pooled with $2 \times 2$ or $3 \times 3$ kernels to shrink the feature map. Following JGMM training and explaining on the pooled features.

**Hyper-parameters.** The training of JGMM requires two hyper-parameters: the number of components of the lower and the higher features. The component number of the lower feature is chosen by validation performance on the validation dataset (split from the training data). When the higher feature is the classification output of the model, the number of components is set to the number of classes; otherwise, the higher feature has already been modelled by a JGMM where it's the lower feature, and the number of component is inherited from the last JGMM. The effective receptive field size is as large as the computation resource allows to capture as many positional information as possible. But when $|R|$ is too large, the training of JGMM will be unacceptably slowed down. We set the effective receptive field size from $2 \times 2$ to $5 \times 5$.

## 4.2 Experimental Settings

**Networks and data.** The experiments are performed on VGG-16/19(Simonyan & Zisserman (2015)), ResNet-18/50(He et al. (2016)) and MobileNet-V1/V2(Sandler et al. (2018)), respectively trained on MNIST (LeCun et al. (1998)), CIFAR-10(Krizhevsky (2009)) and CUB-200(Wah et al. (2011)). The dataset split is the same as the official split. We choose three feature maps from each network, named as 'high', 'middle' and 'low' in the tables and graphics below. In experiments, we train JGMM with 32 components on middle features and 128 components on low features for MNIST; 64 components on middle features and 256 components on low features for CIFAR-10; 128 components on middle features and 1024 components on low features for CUB-200.

**Faithfulness evaluation.** We quantitatively evaluate the faithfulness of JGMM and other interpretable proxy models by classification accuracy on the test dataset (note that the ground truth label for this evaluation is the prediction of the black-box model). As equation 3 shows, JGMM can predict higher feature latent variable distribution given observation of the lower feature. The GMM of the higher feature is dependent on JGMM, which is a biased prediction target, so we only choose the 'high' feature map (which is associated with black-box model classification output) for faithfulness evaluation. By associating each component of the 'high' feature neuron with a dataset class, the JGMM between the 'middle' and the 'high' layers can directly predict class label with uncertainty. The counterpart interpretable proxy models include decision tree (DT), logistic regression (LR), k-nearest neighbor (kNN) and Gaussian process classifier (GP).

**Counterfactuals quality.** The quality of counterfacutal examples is evaluated by the ratio of prediction change from the query input (higher is better) and the distance between the query input and the generated data (lower is better). To mitigate the conflict between the two metrics, we set different constraints on the $l_2$ distance between query feature and the generated counterfacutals. For any tested methods, a generated counterfacutal is rejected if it breaks the constraint. The examples in the test dataset are queried, and we randomly sample 1000 data points by the local explanatory example sampling method in section 3.2. JGMM is compared with MADWachter et al. (2017), DiCEMothilal et al. (2020), Proto-GuidedLooveren & Klaise (2021) and K-d TreeGoyal et al. (2019). The evaluation metric is the counterfactual effectiveness, i.e. the ratio of generated counterfactual examples that have different black-box prediction from the queried example, under a specific distance constraint.

### 4.3 GLOBAL EXPLANATIONS

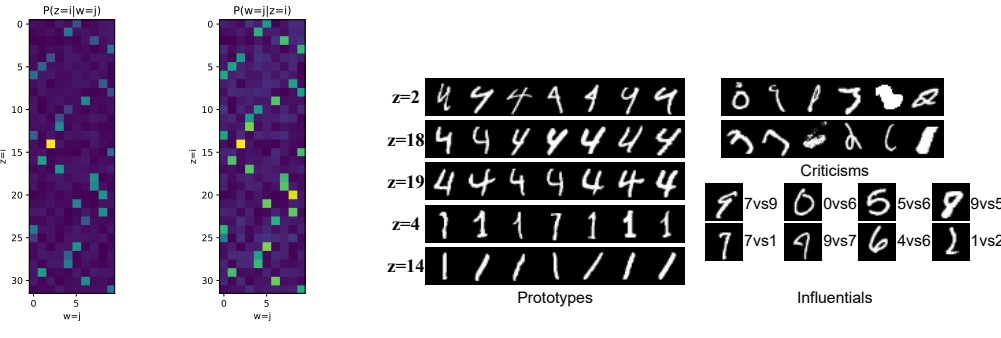

(a) Estimated posterior **Q** and **Ψ**

(b) Global explanatory examples

Figure 4: Global explaining for VGG-19 trained on MNIST. (a) visualizes the posterior distribution of the latent variable, which interprets how JGMM makes predictions. (b) presents global explanatory examples: prototypes (with its component index), criticisms and influentials (with its two most likely labels).

Table 1: Proxy model accuracy (%) evaluation on 'middle' and 'high' layer features with various benchmarks and networks.

| Data | Network | Proxy Model | | | | |
|------|---------|------|------|------|------|------|
| | | DT | LR | GP | kNN | JGMM |
| MNIST | VGG-16 | 85.65 | 93.84 | 94.52 | 89.63 | 97.65 |
| MNIST | VGG-19 | 89.43 | 89.91 | 93.59 | 92.43 | 98.13 |
| CIFAR-10 | ResNet-18 | 88.62 | 89.12 | 91.13 | 83.75 | 96.42 |
| CIFAR-10 | ResNet-50 | 82.31 | 90.45 | 93.73 | 90.89 | 98.01 |
| CUB-200 | MobileNet-V1 | 40.47 | 56.42 | 81.34 | 40.91 | 90.44 |
| CUB-200 | MobileNet-V2 | 30.93 | 69.34 | 85.92 | 64.95 | 94.10 |

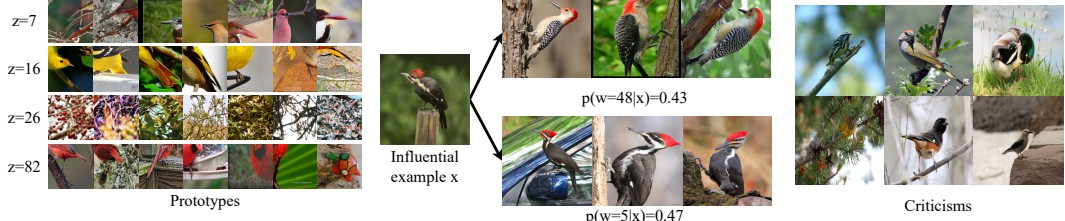

Figure 5: Global explaining for ResNet-18 trained on CUB-200.

Table 1 presents proxy model accuracy of JGMM and other interpretable proxy models on various datasets and networks. It shows that JGMM is highly faithful to the black-box, which increases the credibility of the explanations. We also find that, if the black-box classifier is tasked with more classes, the accuracy of proxy models is lower. This is mainly because the black-box model tends to encode more complex deep features, but the fitting efficiency of interpretable models is often not comparable with the deep network.

Figure 4 presents the estimated posterior **Q** and the derived **Ψ** of JGMM and global explanatory examples mined from the test dataset. The visualization of **Q** suggests that each latent component **z** of the lower feature is mainly mapped to a single higher feature component (though noisy). The visualization of **Ψ** illustrates that each higher feature component (linked to digit class label) has multiple patterns of examples. As observed from **Ψ**, we can find that components $z \in \{2, 18, 19\}$ are learned patterns for digit "4", and $z \in \{4, 14\}$ are learned patterns for digit "1". The prototypes

of those components are visualized. It can be found that each latent component $\mathbf{z}$ is encoded with a certain style of writing. The mined criticisms in figure 4 can hardly be represented by any latent component, which is intuitively true by the examples and technically evaluated by equation 5. The influentials are presented with two most likely $p(w = j|\mathbf{x})$ in equation 6. For example, the first influential example can probably be classified as '7' and '9' with similar posterior probability of $w$.

Figure 5 presents global explanatory examples mined from ResNet-18 trained on CUB-200 with bird species classification task. The 'middle' feature map has encoded different visual concepts into the deep representations, which is revealed by the prototypes, such as $z = 7$ 'long pointed beak', $z = 16$ 'yellow body part', $z = 26$ 'branches texture' and $z = 82$ 'red body part'. An influential example $x$ has close posterior probability on $w = 5$ and $w = 48$. The bird in the influential example has red feather at the top of the head and pointed beak, but the shapes of the red feather and the beak are not typical in either $w = 5$ or $w = 48$, which makes it near the decision boundary. Criticisms of CUB-200 are often rarely observed sub-species or whose image features are not sufficient for discrimination.

## 4.4 LOCAL EXPLANATIONS

Table 2: Counterfactual effectiveness (%) evaluation results with varying $l_2$ constraint ($\sigma$). The experiment is performed on MobileNet-v2 trained on CUB-200.

| Counterfactual Generator | $l_2$ constraint | | | | |
|---|---|---|---|---|---|
| | $\sigma = 0.1$ | $\sigma = 0.5$ | $\sigma = 1.0$ | $\sigma = 2.0$ | $\sigma = 3.0$ |
| MAD Wachter et al. (2017) | 21.68 | 24.73 | 52.41 | 55.91 | 84.22 |
| DiCE Mothilal et al. (2020) | 43.67 | 47.88 | 51.35 | 71.67 | 98.41 |
| Proto-Guided Looveren & Klaise (2021) | 31.95 | 34.53 | 49.96 | 69.01 | 93.64 |
| K-d Tree Goyal et al. (2019) | 52.78 | 69.00 | 78.82 | 81.23 | 98.94 |
| JGMM (ours) | 78.41 | 85.67 | 92.36 | 97.19 | 99.17 |

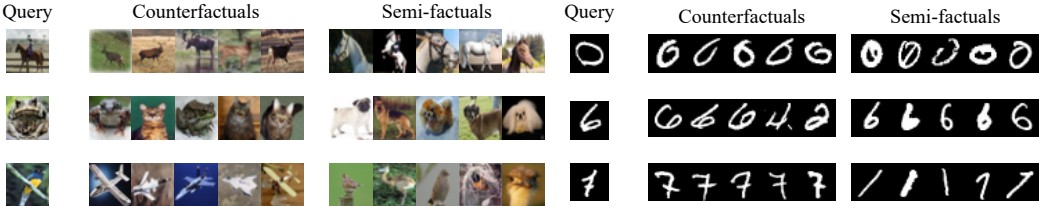

Figure 6: Mined counterfactual and semi-factuals for queries from CIFAR-10 and MNIST.

In table 2, we test the counterfactual generation effectiveness of JGMM and other methods on MobileNet-v2 trained on CUB-200, with $l_2$ constraint of feature distance varying from 0.1 to 3.0. JGMM's sampling method shows higher counterfactual generation effectiveness than the counterparts, especially in low distance-constraint settings. An advantage of JGMM on counterfactual effectiveness is that we can estimate the distance between the generated example and the queried example and reject too distant examples by equation 7, which significantly reduce the number of distant examples.

Figure 6 presents query images for CIFAR-10 and MNIST as well as some mined counterfacutals and semi-factuals. For example, the first CIFAR-10 image query is a side view of a horse, but many deer images have very similar visual features as the query. So most mined counterfacutals are deers viewed from one side, which has different label prediction as the query. The third CIFAR-10 query is a flying bird viewed from the top, which can be easily confused (similar deep features) with a plane, such as the counterfacutals. The semi-factuals provide examples of the same prediction but have very different deep features from the query. The second query of MNIST also demonstrates how the perturbation of deep features will change the prediction. The query image (the black-box model and JGMM predict '6') can be confused with '4' counterfacutals if the last draw is slightly longer. In contrast, other '6' semi-factuals have very different latent encoding.

## 5 CONCLUSIONS

In this paper, joint Gaussian mixture model is proposed to explain the deep features and model inference of black-box DCNN. As a probabilistic model that links two deep features, JGMM enable consistent and diverse explanation forms, including proxy models and example-based explaining. Experiments show that JGMM can faithfully interpret model inference and efficiently mine or generate explanatory examples. The computational efficiency of JGMM worth improvement, especially with large dataset and higher feature dimensionality, as training with EM algorithm in such scenarios is still very time and memory costly.

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

## A  EM ALGORITHM FOR JGMM

The maximum likelihood estimation (MLE) problem of JGMM is the optimization on the log-likelihood of $N$ observations $\{\mathbf{x}_1, \mathbf{y}_1, ..., \mathbf{x}_N, \mathbf{y}_N\}$:

$$\max_{\theta} \sum_{n=1}^{N} log(\sum_{i=1}^{K_x} \sum_{j=1}^{K_y} G(x; \mu_i, \Sigma_i) G(y; \phi_j, \Theta_j) Q_{i,j} \pi_j) \tag{12}$$

$$\tag{13}$$

Let the parameters at the $t$-th iteration step be:

$$\theta^{(t)} = \{\mu_i^{(t)}, \Sigma_i^{(t)}, \phi_j^{(t)}, \Theta_j^{(t)}, \mathbf{Q}_{i,j}^{(t)}, \pi_j^{(t)} | 1 \le i \le K_x, 1 \le j \le K_y\} \tag{14}$$

The expectation step (E-step) involves the estimation of the latent variables based on $\theta^{(t)}$. Let $g_{n,i}^{(t)}$ and $h_{n,j}^{(t)}$ respectively be the expectation of the latent variable $p(z_n = i)$ and $p(w_n = j)$ at the $t$-th

iteration. They computed by:

$$g_{n,i}^{(t)} = \frac{\mathbf{Q}_{i,*}^{(t)} \pi^{(t)} G(\mathbf{x}_n | \mu_i^{(t)}, \mathbf{\Sigma}_i^{(t)})}{\sum_{r=1}^{K_x} \mathbf{Q}_{r,*}^{(t)} \pi^{(t)} G(\mathbf{x}_n | \mu_r^{(t)}, \mathbf{\Sigma}_r^{(t)})}$$

$$h_{n,j}^{(t)} = \frac{\pi_j^{(t)} G(\mathbf{y}_n | \phi_j^{(t)}, \mathbf{\Theta}_j^{(t)})}{\sum_{r=1}^{K_y} \pi_j^{(t)} G(\mathbf{y}_n | \phi_r^{(t)}, \mathbf{\Theta}_r^{(t)})}$$

$$1 \le i \le K_x, 1 \le j \le K_y \tag{15}$$

The maximization step (M-step) maximize the likelihood function by updating the estimation $\theta^{(t)}$ to $\theta^{(t+1)}$. The update of Gaussian parameters is:

$$\mu_i^{(t+1)} = \frac{\sum_{n=1}^{N} g_{n,i}^{(t)} x_n}{\sum_{n=1}^{N} g_{n,i}^{(t)}} \tag{16}$$

$$\mathbf{\Sigma}_i^{(t+1)} = \frac{\sum_{n=1}^{N} g_{n,i}^{(t)} (x_n - \mu_i^{(t+1)})(x_n - \mu_i^{(t+1)})^T}{\sum_{n=1}^{N} g_{n,i}^{(t)}} \tag{17}$$

$$\phi_j^{(t+1)} = \frac{\sum_{n=1}^{N} h_{n,j}^{(t)} y_n}{\sum_{n=1}^{N} h_{n,j}^{(t)}} \tag{18}$$

$$\mathbf{\Theta}_j^{(t+1)} = \frac{\sum_{n=1}^{N} h_{n,j}^{(t)} (y_n - \phi_j^{(t+1)})(y_n - \phi_j^{(t+1)})^T}{\sum_{n=1}^{N} h_{n,j}^{(t)}} \tag{19}$$

The optimization of latent distribution parameters $\mathbf{Q}$ and $\pi$ are:

$$\pi_j^{(t+1)} = \frac{1}{N} \sum_{n=1}^{N} \sum_{i=1}^{K_x} g_{n,i}^{(t+1)} h_{n,j}^{(t+1)} \tag{20}$$

$$\mathbf{Q}_{i,j}^{(t+1)} = \frac{1}{\pi_j^{(t+1)}} \frac{1}{N} \sum_{n=1}^{N} g_{n,i}^{(t+1)} h_{n,j}^{(t+1)} \tag{21}$$

As above, the E-step and the M-step of the EM algorithm for JGMM are defined. We first initialize $\mathbf{Q}$ and $\pi$ by uniform distribution parameters. Then, we cluster the data by K-means ($K_x$ clusters for $\mathbf{X}$ and $K_y$ clusters for $\mathbf{Y}$) and initialize Gaussian parameters by cluster sample mean and covariance. Last, the algorithm execute E-step and M-step until convergence.

