# OpenReview forum: "Joint Gaussian Mixture Model for Versatile Deep Visual Model Explanation"
_ICLR.cc/2023/Conference — Submitted to ICLR 2023_

### Official Review · Reviewer_XcXy · 2022-10-22

**Confidence:** 3
**Correctness:** 3
**Technical Novelty And Significance:** 3
**Empirical Novelty And Significance:** 3
**Recommendation:** 8

**Clarity, Quality, Novelty And Reproducibility:**

Overall, the paper is well-written. The presented method is believed novel. The reproducibility is likely OK as no code is available during the review.

**Strength And Weaknesses:**

Strength.
(1) The paper is well-written with clear logic.
(2) The presented method is sound.
(3) The empirical experiments are convincing.

Weaknesses.
(1) The limit of the proposed JGMM is not fully demonstrated, e.g., under what circumstances will JGMM not work well?
(2) Section 3.3 is believed crucial for improving the practicability of the JGMM. However, this section is not easy to follow.

**Summary Of The Paper:**

A new a probabilistic model, named joint Gaussian mixture model(JGMM), is proposed for post-hoc explanations of deep neural networks. By jointly modeling the latent features of lower and higher layers with a joint GMM, JGMM, trained with a post-hoc EM algorithm, is capable of delivering both global explanations (like Prototypes, Criticisms, and Influentials) and local explanations (like counterfactual and semifactual examples). Experiments on common networks and datasets are conducted to demonstrate the effectiveness of the proposed method.

**Summary Of The Review:**

Explanations of deep neural networks are essential. The presented JGMM might serve as an explaining framework that produces versatile, consistent, faithful, and understandable explanations.

Notations should be revised carefully. For example, those used in the figures are inconsistent with those in the text. In Eq (1), the function G() is not defined.

I am quite concerned about using the GMM to model lower features (like the input images). Is GMM expressive enough? Or how does Section 3.3 help with that?

---

> ### Author Response · Authors · 2022-11-18
> **Reply to reviewer XcXy(#4)**
>
> Thank you for your careful and encouraging review. We summarize your major concerns or questions about this paper as below. Our replies follow.
>
> (1)	The limit of JGMM. The major limit of JGMM is the training efficiency. JGMM is trained with EM algorithm in an iterative manner. Our implementation shows training JGMM with large or high-dimensional dataset is very slow and memory costly. This is why we train JGMM on CUB-200 with sub-sampling on the deep features. The inefficiency problem restricts JGMM's potential of application on more complex model or larger image data.
>
> (2)	The understandability of section 3.3. It's also suggested by other reviewers to improve the writing of this section. We think the most difficult concept is the relative receptive field $R$. So we introduce $R$ more carefully in the revised version.
>
> (3)	$G$ in Eq(1). $G$ is actually Gaussian distribution density function. We omitted the Introduction of this notation in the first version, but it's fixed in the revised version.
>
> (4)	Modelling lower features by GMM. We failed to model very low feature maps, because of the training inefficiency problem stated in (1). The lower feature maps are very large, which makes EM learning unendurably slow.

---

### Official Review · Reviewer_DQ8v · 2022-10-23

**Confidence:** 5
**Correctness:** 3
**Technical Novelty And Significance:** 4
**Empirical Novelty And Significance:** 3
**Recommendation:** 8

**Clarity, Quality, Novelty And Reproducibility:**

Clarity: good; Quality: good; Novelty: good;
For the reproducibility, there is still room for improvement. More experimental details should be clarified.

**Strength And Weaknesses:**

[Strength]
1. This paper pointed out a potential problem in Explainable AI, that is, different types of explanations were produced from different explanation methods/systems, which may result in incompatibility and inconsistency between these explanations. To address the problem, the authors proposed a probabilistic JGMM framework that can simultaneously produce various types of explanations.
2. Experiments verified the effectiveness of the proposed JGMM framework.
3. This paper is clearly written, and easy to understand.

[Weakness]
1. This paper was motivated by a potential problem in XAI, i.e., different types of explanations may be incompatible and inconsistent with each other. However, no theoretical or empirical evidences were provided to support such a claim. It would be great if the authors can give an example to show how different types of explanations will conflict with each other in a specific task.
2. More experiments are encouraged to demonstrate the effectiveness of the proposed JGMM framework.
a)	For the faithfulness evaluation in Table 1, the authors only compared JGMM with four traditional proxy methods. However, there are many other interpretable models with better performance. A convincing demonstration should include a comparison with SOTA interpretable proxy models.
b)	For many types of explanations like prototypes and criticisms, The authors only showed explanation results produced by the proposed JGMM. There lacks a comparison between the proposed JGMM and corresponding baseline methods. It would be nice to see such a comparison, to show whether the proposed JGMM produced better explanation results or not.
c)	Some additional experiments were needed to further verify the consistency among different types of explanations produced by the proposed JGMM.
3. The proxy model accuracy in Table 1 used “middle features” for prediction, why not use “low features”? If the proxy model accuracy using “low features” was poor (i.e., low features may not be faithful), were other types of explanations based on “low features” faithful?

Minors

4. In experiments, the authors adopted “high feature”, “middle feature”, and “low feature”. What is the exact definition of “middle feature”?

5. In Table 2, the evaluation metric “Counter-factual effectiveness” is not clearly defined in the paper.

6. In Equation (2), the summation in the denominator should be from k=1 to k=K_y, rather than K_x.

7. In Section 2, it is not appropriate to refer to LIME as a proxy model.

8. It would be more clear to give a table to list hyper-parameters used in the proposed JGMM.


**Summary Of The Paper:**

This paper proposed a probabilistic joint Gaussian mixture model (JGMM) that could simultaneously produce various forms of explanations for DCNN, including proxy models, prototypes, criticisms, influential examples, counter-factual examples and semi-factual examples. In this way, the compatibility and consistency among different explanations could be guaranteed. Experiments were conducted to demonstrate the effectiveness of the proposed JGMM framework.

**Summary Of The Review:**

This paper is interesting and has significant contributions. It pointed out a potential problem in Explainable AI, i.e., different types of explanations were produced from different explanation methods/systems, which may result in incompatibility and inconsistency between these explanations. To address the problem, the authors proposed a probabilistic JGMM framework that can simultaneously produce various types of explanations. Experiments verified the effectiveness of the proposed JGMM framework. Besides, this paper is clearly written, and easy to understand. Nevertheless, I still have some concerns about this paper. Please refer to the weakness in my review. It would be great if my main concerns can be well addressed.
In summary, I would suggest to accept this paper.

---

> ### Author Response · Authors · 2022-11-18
> **Rely to reviewer DQ8v(#3)**
>
> Thank you for your valuable comments. We are pleased to learn your recognition on the value of the versatility and consistency of model explaining.
>
> We reply to each of your concerns as below:
>
> 1. Unfortunately, it's difficult to provide solid evidence in this paper for the potential incompatibility and inconsistency problem of XAI. We raise this problem in this paper because any explanation is unavoidably biased and there's risk of conflicting bias. JGMM is a workaround to this problem.
>
> 2. We understand that extensive experiments are important. But a major limit of JGMM is that training with high-dimensional or large-size data is very inefficient. JGMM is trained with two GMMs by EM algorithm, which is very slow and memory costly. We think it's more imperative to improve the training efficiency of JGMM before larger scale experimentation.
>
> 3. It's stated in section 3.2 that, because the higher features are actually the classification output of the DCNN models, which has discrete labels and can be leveraged as a classification problem. In contrast, middle and low features are continuous values.
>
> 4. "middle feature" is the feature map of a layer between that of "high" layer and "low" layer. For example, for ResNet-18, the low feature is the feature map of the 2rd residual layer, and the middle feature is the feature map of the 4-th residual layer.
>
> 5. The explanation of this metric is added to section 4.2.
>
> 6. The typo is corrected in the revised version.
>
> 7. We understand that LIME is a local surrogate originally proposed to explain the sensitivity of features locally. But it can also work as a linear local proxy model to predict black-box predictions.
>
> 8. Because a table will be spacious and the room for main paper is very limited, the hyper-parameter settings are updated in section 4.1 and 4.2.

---

### Official Review · Reviewer_gUvR · 2022-10-23

**Confidence:** 5
**Correctness:** 1
**Technical Novelty And Significance:** 2
**Empirical Novelty And Significance:** 1
**Recommendation:** 3

**Clarity, Quality, Novelty And Reproducibility:**

The authors mention that most existing ‘DCNN explaining methods are single-purpose systems’. Note that a proxy model like LIME can provide local explanations, but it is also able to provide counterfactuals if needed (one simply needs to use gradient-based methods on the locally linear proxy model to obtain such counterfactuals).

The PGM in Figure 2 is a bit confusing, especially the lower plate. It does not look like a typical PGM, or a factor graph representation of a probabilistic model. For example, I assume that Sigma and mu should both directly point to x?

Empirically, I can see what the authors mean to design by using the approximate posterior $q_k$ as the prior for $z_n$. However, the model is not principled; that is, it does not seem to be a rigorous probabilistic graphical model (PGM) formulation any more. In this case, it might be better not to describe the model as a PGM, but directly introduce the method as a computational graph. The authors should also revise Sec. 3.1 such that the description on prior and posterior is more rigorous. According to Eq (1), it seems Q is just a global parameter matrix that is related to both GMMs. In this case, the claim that the prior of z depends on the posterior of w is rather confusing.

The authors refer to having only access to intermediate layers as a ‘black-box’ setting. This is somewhat inaccurate. In a black-box setting, one only has access to input and output of the model, but not the intermediate features. To me, this is more like the gray-box setting used in [1].

Another one of my major concerns is the limited technical merit. The proposed method is a direct application of GMM on the features X and Y. The only difference is that, instead of having one GMM modeling X, the authors propose to use two GMMs, one for X and one for Y, and connect them with a projection matrix Q. The idea of joint modeling is not new either [2].

The learning algorithm also follows the typical EM algorithm for learning GMMs.

There are also places in the derivation that may not be correct. For example, the set of equations above Eq (2) is a bit confusing. Why is $[g_x]_i$ a Gaussian distribution parameterized by ($\hat{\mu}_i, \hat{\Sigma}_i$)? Shouldn’t it be one entry of the vector $g_x$? What parameters are $\Theta$ and $\lambda$? In other words, what posterior distributions are $\Theta$ and $\lambda$ associated with? What is the iterative algorithm to estimate these parameters. For example, how is $Q$ estimated? Without these important details, it is difficult to evaluate the correctness of the paper’s method part. The authors need to move more details from the Appendix to the main paper. The main paper in its current form still needs some work.

Equation (4) is confusing and potentially problematic. The authors mention that this refers to directly sampling from a multivariate Gaussian distribution $p(x| w = j)$. However, $w$ is the component ID for the second GMM for $y$, while $x$ is model by the first GMM. From Sec. 3.1, if one uses $q_j$, the sampling is in fact also from a GMM, not from a Gaussian distribution.

It is unclear how the definition of counterfactuals in Eq (7) is consistent with the standard definition of counterfactuals in Pearl’s do-calculus. It is also unclear how $\alpha_i$ in Eq (8) is related to $p(z=i|x)$ in Eq (7).

The extension of the proposed method to DCNN is unclear. Specifically, the authors proposed to train $|R|$ matrices of $Q$ between S and T, but from the text it is unclear which $|R|$ positions to choose.

Eq (9) is incomplete. It is unclear why the definition is the reverse of counterfactuals.

Given that one of the focus in evaluation is to generate counterfactual explanations, some important baselines are missing, e.g., [3].

The baselines (decision trees, logistic regression, etc.) used for evaluating proxy model accuracy are weak. The authors did not even include the 2016 method LIME as a baseline. More recent baselines should be included. Otherwise it is difficult to evaluate whether the proposed method is actually doing well in terms of proxy model accuracy.

In Figure 5, for the influential example case, it is hard to know what shape of red feather and beaks are typical. It would be helpful to provide more ‘typical’ images rather than only one of them.

[1] Training-Free Uncertainty Estimation for Dense Regression: Sensitivity as a Surrogate. AAAI 2022.

[2] Speaker Adaptive Joint Training of Gaussian Mixture Models and Bottleneck Features. ASRU 2022.

[3] CounteRGAN: Generating Counterfactuals for Real-Time Recourse and Interpretability using Residual GANs. UAI 2022.

Minor:

There should be a ‘space’ before every bracket.


**Strength And Weaknesses:**

Strength

+ Generating model explanations is an important and interesting problem for the ML community.
+ The proposed model can generate a wide range of explanation types including prototypes and counterfactuals.
+ The definitions of prototypes, criticisms, etc. make sense to me, though (1) it is unclear how the definition of counterfactuals in Eq (7) is consistent with the standard definition of counterfactuals in Pearl’s do-calculus and (2) since Eq (9) is incomplete, it is unclear why the definition is the reverse of counterfactuals.

Weaknesses

- The proposed model is not principled; that is, it does not seem to be a rigorous probabilistic graphical model (PGM) formulation any more.
- A lot of details on Equation (2) and (3) are missing/unclear in the main paper.
- Equation (4) is confusing and potentially problematic.
- Given that one of the focus in evaluation is to generate counterfactual explanations, some important baselines are missing, e.g., [3].
- The baselines (decision trees, logistic regression, etc.) used for evaluating proxy model accuracy are weak. The authors did not even include the 2016 method LIME as a baseline.


**Summary Of The Paper:**

This paper proposed a simple joint Gaussian mixture model to generate post-hoc model explanations for DCNN. Essentially the authors use two GMMs to model the higher and lower features Y and X, with a projection matrix Q connecting the component probabilities from these two GMMs. Experiments demonstrated the proposed method’s capability of generating local and global model explanations.

**Summary Of The Review:**

This paper proposed a simple joint Gaussian mixture model to generate post-hoc model explanations for DCNN. Essentially the authors use two GMMs to model the higher and lower features Y and X, with a projection matrix Q connecting the component probabilities from these two GMMs.

Overall I feel the problem the paper explores, i.e., to develop a probabilistic model that is capable of making versatile explanations, is interesting. One of my major concerns is the limited technical merit of the proposed method. Besides, the chosen baselines are severely out of date, especially for the global part (Table 1). Another major concerns is that there are various inaccuracies and missing details that make one question the correctness and rigor of the paper. I feel these are difficult, if not impossible, to address in a single round of conference revision at ICLR.

---

> ### Author Response · Authors · 2022-11-18
> **Reply to reviewer gUvR(#2)**
>
> Thank you for your insightful and valuable comments on this work. To summarize, your major concerns with this paper includes the following aspects. Our answers are presented to address your concern or acknowledge the mistakes or the limits of our work.
>
> Problem 1: The correctness and clarification of the probabilistic model, the math notations, and the derivation.
>
> Answer 1: We respectively respond to the potential problem with each part that you have pointed out.
>
> Eq(2) & Eq(3): We argue that every symbol in Eq(2) and E1(3) is defined. The referred variables, e.g. $\hat{Q}$ and $\mu$, are estimated JGMM parameters by MLE, as clarified in the paper. We define the symbols $g_x$, $g_y$, $\hat{\lambda}$, $\hat{\Psi}$ in Eq(2) as shortcuts to some posteriors used in Eq(3).
>
> Eq(4): In Eq(4), we are interested in the distribution of lower feature $X$ given a higher feature component $w=j$, thus we sample from $p(x|w=j)$ instead of $p(x|z=i)$. $p(x|w=j)$ is a GMM because $p(z|w=j)$ can be computed by $\hat{q}_j$, and the resulted distribution of $X$ is a GMM whose component weight vector is $\hat{q}_j$.
>
> Eq(4) is the density function of a GMM parameterized with the estimated $\hat{\mu}$, $\hat{Sigma}$, $\hat{q_j}$. The mean points of the Gaussian components are naturally clustering centroid, which is regarded as the prototypes of the clusters.
>
> Fig(2): Fig(2) is produced by a LaTeX package(https://github.com/jluttine/tikz-bayesnet). The style might be unfamiliar, but the graph is correct in the presented style. Probably it's the dashed line boxes that have confused you. Each box means a named distribution ($N$ stands for Normal distribution, $Cat$ stands for a discrete categorical distribution), while the lines are linked to distribution parameters. For example, $\Sigma_i$ and $\mu_i$ are parameters of the left normal distribution, so they are linked to the box at the left-side.
>
> Eq(7):  Eq(7) is actually the expansion of $p(z=z_{query}, w\neq w_{query}|x, x_{query})$. $z=z_{query}$ is to restrict that $x$ is not too distant from $x_{query}$, i.e. it's likely that $x$ and $x_query$ are sampled from the same component; but $w\neq w_{query}$ encourages the difference of the decision.
>
> Eq(9): There's a typo with Eq(9). It's fixed in the revised version.
>
> Figure (3): $R$ is a hyper-parameter of the extended JGMM. It's the relative receptive field of the higher layer on the lower layer.
>
> Your other confusions about section 3.1: Note that in Eq(2) and Eq(3), $x$ and $y$ are observations(described in the text before Eq(2)). So $[g_x]_i$ is not a Gaussian distribution. It's the evaluated density function at point $x$, and so as $[g_y]_j$. $\Theta$ is associated with the GMM on $Y$. $\Theta$ is updated by Eq(19). $\hat{\lambda}$(Eq(2)) is not a parameter in JGMM. It's just a shortcut to the estimated posterior $z$. Your suggestion of moving the training algorithm from the appendix to the main paper could improve the readability and understandability of Section 3.1. But, fortunately, the main paper would exceed the page limit if we do so.
>
> Problem 2: JGMM is not principled.
>
> We argue that JGMM is a probabilistic graphical model. It's a variant of GMM by jointly modelling on two features, in which GMM its-self is a PGM. We do not define JGMM as a computational graph because it represents a parametric probabilistic model and the parameters can be estimated by classical EM algorithm.
>
> Actually, $z$ and $w$ are mutually dependent. When $X$ is observed, posterior of $Z$ can be estimated; through the connection provided by $Q$, posterior of $w$ can also be estimated. The reverse estimation from observe $Y$ to the estimated $z$ also holds.
>
> Problem 2: Comparison with other proxy model baselines.
>
> Answer 2: The reason why the mentioned models are not involved in our experiment as counterparts: (1) Local proxy models like LIME are not compared, because they are not capable of being trained as global proxy models. (2) We do not compare JGMM with generative models because JGMM find counterfactual (as well as other explanatory examples) by searching, not generation.
>
> Problem 3: Novelty.
>
> JGMM is completely different from [1]. In [1], it is the DNN features are jointly trained with GMM; in JGMM, the deep features are freezed, and the models jointly trained are two GMM connected by $Q$.
>
> [1] Speaker Adaptive Joint Training of Gaussian Mixture models and Bottleneck Features.

---

> > ### Comment · Reviewer_gUvR · 2022-12-11
> > **Thank you for your response**
> >
> > I have read the author response and other reviews. I would like to thank the authors for their response, which is helpful in clarifying some of the confusions. However, my concerns on the gray-box setting along with PGM formulation are not very well addressed. I also agree with some of the points from Reviewer pdiK, e.g., efficiency evaluation of the method and major concerns on presentation of the methodology. BTW, the current manuscript has exceed the limit of 9 pages.
> >
> > I therefore would like to keep my score unchanged.

---

### Official Review · Reviewer_pdiK · 2022-10-24

**Confidence:** 4
**Correctness:** 3
**Technical Novelty And Significance:** 4
**Empirical Novelty And Significance:** 3
**Recommendation:** 5

**Clarity, Quality, Novelty And Reproducibility:**

As mentioned in the Weaknesses section, the paper is not clarified clearly, including methodology and experiments, although the proposed method is novel and technically sound. There are many typos and mistakes, and the total quality of the paper should be improved. The reproducibility is therefore not good.

**Strength And Weaknesses:**

Strength:
1. The authors proposed a novel viewpoint of DNN model explanation by mixture models, which introduces the correlation between the features of two layers into the probabilistic model instead of independently modeling in some other methods. The joint modeling can explain the learned representations of higher/lower-level features as well as the black-box model between them.
2. The JGMM is applicable to any black-box DNN model as a post-hoc interpreter.
3. It can produce explanations in various image classification tasks, which has been shown in the experiments. The visualizations can clearly show their superiority. Various forms of model explanations can be efficiently produced from the framework. The consistency among different explanations is also demonstrated.

Weaknesses:
1. The writing of this paper is not good enough. Some sentences are hard to understand, e.g., “a probabilistic model jointly models inter-layer deep features and produces faithful and consistent post-hoc explanations” in abstract, “The deep feature maps of DCNN usually has spatial dimensions” in section 3.3.
2. The position of variable Q is not clarified clearly. How to understand it under a probabilistic view? Besides, it is hard to understand how it works. Eq (22) shows its updating function, are there any references that can support it?
3. The paper has many typos, e.g., \hat{q} in eq (4) is undefined, a variable i in eq (9) is lost, many variables are unclearly defined. The authors should carefully check their paper.
4. How to select the component number of the two GMM? How to select which two layers are used for modeling the JGMM? Do different values affect the results? Are there any ablation studies for clarifying the points?
5. The definitions of global and local explanations should be clarified.
6. The comparisons of complexity and parameter number between JGMM and others should be discussed in the paper.



**Summary Of The Paper:**

In this paper, the authors proposed a joint Gaussian mixture model (JGMM)-based post-hoc explanation method, which applies inter-layer deep features in a probabilistic model. The JGMM can explain deep features and inter-layer deep feature relationship on the latent component variables in its GMMs. The JGMM is applicable to any black-box DNN model for providing explanations for higher/lower level features and the black-box model between them.

The proposed method is interesting and technically sound. It can obtain different forms of model explanations with same trends. However, the writing of this paper is poor which should be improved by the authors. In addition, the complexity of the JGMM should be discussed.


**Summary Of The Review:**

As described above, the authors should carefully check and revise their paper. The whole paper should be improved.

---

> ### Author Response · Authors · 2022-11-18
> **Reply to Reviewer pdiK(#1)**
>
> Thank you for your careful and thoughtful feedback. We are pleased to know your recognition of the novelty and the value of this paper. As your comments suggest, the paper needs to be improved by clarification on some of the important concepts and fixing misleading language flaws. Our response to your concern or questions is as below. It's also reflected in the updated version of the paper.
>
> Weakness 1: The paper is carefully proofread again. We have rephrased some less understandable texts.
>
> Weakness 2: The posterior probability matrix $Q$ and $q_k$ are defined in Section 3.1. $q_k$ is the $k$-th column vector of $Q$, and $Q_{i,j}$ denotes the probability $p(z=i|w=j)$. We use the notation $q_k$ in Figure 2 for consistency with the connection between categorical variables $w_n$ and $z_n$ in the graph.
>
> $Q$ is the conditional probability $p(z|w)$ on different values of discrete variables $z$ and $w$. This means, when $Q$ has been estimated as $\hat{Q}$, the discrete posterior distribution of $z$ can be estimated by any assigned value of $w$; and vice versa by equation (2). It helps exploring the connection between the GMM clusters of the lower and the higher features.
>
> Equation (22) in the original version has been renumbered as Equation (21). Equation (21) is the updating rule of $Q$ in the M-step, derived from the maximum problem (M-step) on $\hat{Q}$.
>
> Weakness 3: $\hat{q_j}$ in Equation (4) is the $j$-th column vector of the estimated posterior $\hat{Q}$. Its definition is appended to where it firstly shows. We have corrected some other typos involved with mathematical notations, including Equation (9).
>
> Weakness 4: The last paragraph of Section 4.1 briefly summarizes how the number of components is chose empirically. The third paragraph of Section 4.1 also provides some rule-of-thumb for the selection of DCNN layers, with the trade-off between data exploitation and the training efficiency of JGMM. However, the selection of the hyper-parameters is still totally empirical.
>
> We haven't performed extensive experiment on the tuning of the hyper-parameters due to the training efficiency of JGMM. Training JGMM with EM algorithm and very large dataset size (note that each position on the feature map is a data point) is very slow and memory costly. We consider that, in the future work, only the improvement on the training efficiency of JGMM will allow more experimental exploration.
>
> Weakness 5: The concepts global and local explanations are explained in the fourth and fifth paragraphs of 1. Introduction. In a nutshell, global explanations reveal model behaviours on the whole data distribution, showing the general rules or patterns learned by the model; local explanations on associated with a specific data point, useful for evaluation on the sensitivity of model decision or case-by-case analysis.
>
> Weakness 6: We understand that the computational complexity and the number of parameters will be helpful for the estimation of the training/inference efficiency of JGMM. However, unlike DCNN models, whose forward-propagation speed is very relevant to the number of layers, FLOPS and the number of parameters, we are afraid that the comparison between different proxy models(such as decision trees, linear models) is not as relevant to the realistic speed of training and inference.
>
> For example, the parameters of JGMM include: $K_x$ components of lower feature GMM, in which each component has a $D_x$-D mean vector and a $D_x \times D_x$ covariance matrix; same for the $K_y$ components of the high feature GMM; a $K_x\times K_y$ posterior matrix $Q$. The parameter number of JGMM is not far less than that of a typical DCNN, but the training of JGMM with EM algorithm is significantly slower than DCNN with SGD.

---

> > ### Comment · Reviewer_pdiK · 2022-11-29
> > **Response**
> >
> > Thanks the authors for their respones. I still have some concerns about the paper.
> > For the response to weakness 4, it is acceptable to conduct ablation studies on a small dataset, like CIFAR-10.
> > For the response to weakness 6, although the comparisons of complexity and parameter number may be unfair, the authors should also show their values to clarify the efficiency of the proposed method.
> > I hope the authors can consider the above questions.

---

### Decision · Program_Chairs · 2023-01-20

**Decision:**

Reject

**Justification For Why Not Higher Score:**

The reviews for the paper are mixed. Particularly concerning are reviewers comments that the probabilistic graphical model is not well formulated, and that important baselines are omitted in the experiments section. The reviewers also noted a higher number of grammatical mistakes than average. With the suggested improvements, I believe that the paper will be stronger.

**Justification For Why Not Lower Score:**

N/A

**Metareview: Summary, Strengths And Weaknesses:**

The authors propose a joint GMM model for creating post-hoc explanations for high- and low-level features of a DNN. Better understanding of deep learning models is a very important topic, and it's great that the authors focused on this area. The reviewers, however, pointed out important issues with the model and manuscript: the PGM is not well formulated, setting is incorrectly specified (black-box vs gray-box), the hyperparameter choices not well explained, and the manuscript contains many typos. The paper would improve with a careful reevaluation of these areas.